# An Assassin’s Secret: Multifunctional Cytotoxic Compounds in the Predation Venom of the Assassin Bug *Psytalla horrida* (Reduviidae, Hemiptera)

**DOI:** 10.3390/toxins15040302

**Published:** 2023-04-20

**Authors:** Maike Laura Fischer, Benjamin Fabian, Yannick Pauchet, Natalie Wielsch, Silke Sachse, Andreas Vilcinskas, Heiko Vogel

**Affiliations:** 1Department of Insect Symbiosis, Max Planck Institute for Chemical Ecology, 07745 Jena, Germany; mfischer@ice.mpg.de (M.L.F.);; 2Research Group Olfactory Coding, Max Planck Institute for Chemical Ecology, 07745 Jena, Germany; 3Research Group Mass Spectrometry/Proteomics, Max Planck Institute for Chemical Ecology, 07745 Jena, Germany; 4Institute for Insect Biotechnology, Justus Liebig University, 35392 Giessen, Germany; 5Branch Bioresources of the Fraunhofer Institute for Molecular Biology and Applied Ecology, 35392 Giessen, Germany

**Keywords:** venomics, Reduviidae, cytotoxicity, redulysin, venom protein family 2

## Abstract

Predatory assassin bugs produce venomous saliva that enables them to overwhelm, kill, and pre-digest large prey animals. Venom from the posterior main gland (PMG) of the African assassin bug *Psytalla horrida* has strong cytotoxic effects, but the responsible compounds are yet unknown. Using cation-exchange chromatography, we fractionated PMG extracts from *P. horrida* and screened the fractions for toxicity. Two venom fractions strongly affected insect cell viability, bacterial growth, erythrocyte integrity, and intracellular calcium levels in *Drosophila melanogaster* olfactory sensory neurons. LC-MS/MS analysis revealed that both fractions contained gelsolin, redulysins, S1 family peptidases, and proteins from the uncharacterized venom protein family 2. Synthetic peptides representing the putative lytic domain of redulysins had strong antimicrobial activity against *Escherichia coli* and/or *Bacillus subtilis* but only weak toxicity towards insect or mammalian cells, indicating a primary role in preventing the intake of microbial pathogens. In contrast, a recombinant venom protein family 2 protein significantly reduced insect cell viability but exhibited no antibacterial or hemolytic activity, suggesting that it plays a role in prey overwhelming and killing. The results of our study show that *P. horrida* secretes multiple cytotoxic compounds targeting different organisms to facilitate predation and antimicrobial defense.

## 1. Introduction

Assassin bugs (Reduviidae) comprise a diverse group of hematophagous and zoophagous heteropterans found in terrestrial habitats around the world. Predacious assassin bugs inject toxic saliva to capture their prey [1,2], but they are also known for their painful defensive bites [3,4]. Their venom gland complex is usually subdivided into an anterior and posterior main gland (AMG and PMG, respectively) and an accessory gland (AG) [5,6]. Recent studies have shown that some reduviid species are capable of using venoms from different glands in a context-dependent manner. While only PMG venom is involved in prey envenomations, AMG venom is injected in response to harassment in two species, i.e., the harpactorine bug *Pristhesancus plagipennis* Walker and the reduviine bug *Psytalla horrida* (Stål) [5,7]. Defensive venom is directed primarily against vertebrates and usually causes pain and tissue damage to deter predators [3,4,8]. In contrast, venom used for prey capture must rapidly immobilize and kill prey and facilitate the extra-oral digestion (EOD) of invertebrate tissue [1,2,7,9,10]. In addition, venom likely contributes to the suppression of microbial growth to prevent the ingestion of potential pathogens or a microbial infestation of the venom glands. [11,12]. To fulfill these distinct functions, assassin bugs need complex venoms that are active against a variety of targets.

The AMG and PMG of reduviid bugs secrete distinct and complex protein mixtures, but PMG venom in particular confers potent toxic activity towards different targets [5,7]. PMG venom from *P. horrida* and *Platymeris biguttatus* (L.) was found to be highly active against *Escherichia coli*, erythrocytes, and insect cells, and is able to rapidly paralyze and digest *Galleria mellonella* (L.) larvae [7]. A strong paralyzing and liquefying action of PMG venom has also been demonstrated for *P. plagipennis* [1,5]. In addition, venom collected from *Rhynocoris iracundus* (Poda) was shown to have toxic effects on mammalian cells, bacterial cells, and helminths [12,13]. Defensively propelled PMG venom from *Platymeris rhadamanthus* (Gerstaecker) has major lytic effects on mammalian neuronal cells, leading to rapid calcium influxes [14]. Although the venom composition of several assassin bug species has been analyzed recently [5,7,14], little is known about the compounds responsible for the observed effects. The most abundant peptides in assassin bug venoms include digestive enzymes, protease inhibitors, putative pore-forming proteins and neurotoxins, and a large number of uncharacterized proteins [1,5,7,12,14,15]. A recent study attempted to identify the insecticidal venom components of the red tiger assassin bug *Havinthus rufovarius* (Bergroth) by determining the toxic activity of venom fractions. The fractions with the strongest paralytic and lethal effects on sheep blowflies contained primarily a CUB domain protein and a cystatin, suggesting that one or both proteins contribute to the venom’s insecticidal activity [16]. In the hematophagous bug *Triatoma infestans* (Klug), trialysin has been identified as a major toxic component that exerts cytotoxic effects against bacteria, protozoa, and mammalian cells by disrupting cell membranes. It was shown that upon proteolytic activation, trialysin forms voltage-dependent channels in lipid bilayers, thus permeabilizing cells [11,17]. In predacious reduviids, homologous proteins called redulysins are among the highest expressed proteins in the PMG and likely contribute to the venom’s toxic effects [1,7,12,14]. Moreover, trialysin/redulysin homologs are not only found in predacious true bugs but also in phytophagous species such as *Riptortus pedestris* (Fabricius), where they can act as antimicrobial agents [18]. A protein family that is also very abundant and highly expressed in the venom glands of many predatory bugs analyzed to date, is the heteropteran venom protein family 2 [5,7,14,15,19]. Similar to redulysins, family 2 proteins contain several conserved cysteine residues [19]; however, their function, activity, and role in the predacious lifestyle are still unknown.

Venoms are an important source of novel active molecules that may be used for drug or pesticide development [9]. Despite recent advances in the analysis of heteropteran venoms, the activity, function, and mode of action of most components remain uncharacterized. This study aimed to identify the venom compounds that are responsible for the antimicrobial, hemolytic, and insecticidal effects of PMG venom from the African assassin bug *P. horrida*. We fractionated crude venom extracts and screened the fractions for toxic effects. Using LC-MS/MS, we analyzed the composition of the active protein fractions and selected candidates for peptide synthesis or heterologous expression. We found several proteins that exert toxic activity towards different targets, thus providing novel insights into venom function and activity in assassin bugs.

## 2. Results

### 2.1. Screening of Venom Fractions

Fractionation of PMG extracts using cation-exchange chromatography resulted in 43 protein fractions with partially overlapping banding patterns of different intensities (Figure 1 and Appendix A). Crude PMG venom has strong cytotoxic effects (Appendix A); therefore, the fractions were screened for cytotoxic, antimicrobial, and hemolytic activity. The observed cytotoxic effects were mainly caused by two fractions (fraction A = 26; fraction B = 41; Figure 1 and Appendix A). Fraction A had strong effects on insect cell viability (22% viability; Figure 1) and bacterial growth (33% growth after 12.5 h; Figure 2B), but no significant hemolytic activity (Figure 2C). In contrast, fraction B had no significant effects on cell viability (Figure 2A), but comparable effects on bacterial growth (27% growth after 13 h; Figure 2B) and stronger hemolytic activity (85% erythrocyte integrity; Figure 2C). In addition to cell-culture-based assays, we screened for insecticidal effects on olfactory sensory neurons in the brain of *Drosophila melanogaster* Meigen using calcium imaging (Figure 3A,B). Application of unfractionated PMG venom on the exposed brains led to an alternating decrease–increase response of intracellular calcium levels ([Ca^2+^]_i_) in the antennal lobe. Although both the timing and amplitude of the effects varied between replicates, the general pattern remained the same for all flies tested (Figure 3C). In contrast, the application of fraction A did not lead to the abovementioned pattern but resulted in a quick drop in fluorescence intensity (Figure 3D). The fluorescence intensity after 10 min of scanning was significantly lower in flies treated with fraction A than in those treated with the negative controls (Figure 3E,F). Fraction B triggered similar calcium responses as fraction A, but the results cannot be interpreted with confidence due to strong buffer effects (Appendix A).

### 2.2. Protein Composition of Active Fractions

Proteomic analysis revealed that fractions A and B were highly similar in their composition. Most identified proteins could be assigned to the S1 peptidase family (12 in fraction A, seven in fraction B). Furthermore, we detected redulysins (one in fraction A, one in fraction B), venom protein family 2 proteins (three in fraction A, two in fraction B)) and gelsolins (one in both fractions). Most identified proteins were strongly expressed in the PMG. In both fractions, gelsolin had the highest total score in the LC-MS/MS analysis (Figure 4). Due to their high PMG-specific expression levels and abundance in predacious true bugs, redulysin, and venom protein family 2 were selected as candidate families for further analysis of cytotoxic activity.

### 2.3. Activity of Synthetic Redulysin Peptides

We identified 12 redulysins in the genome of *P. horrida*, seven of which were also present in the PMG transcriptome and six in the proteome (Appendix A). Eleven out of twelve redulysins that were identified in the genome of *P. horrida* contained a conserved motif homologous to the DEER cleavage site in *T. infestans* (8 × DEER, 2 × NEER, 1 × DEQE) [11], which was followed by a lysine-rich region and a region with usually up to eight cysteine residues. The main lytic regions of these redulysins were predicted based on their homology to the lytic region of the characterized *T. infestans* trialysin (Genbank accession: AAL82381.1) [17] and synthesized by solid-phase synthesis (Appendix A). The two redulysins g1037.t1 and g1038.t1 differed from other isoforms by their exceptionally high molecular weight (Appendix A). In both proteins, we found several acidic regions with potential cleavage sites (4 × GILK, 3 × DEEK, 2 × DEEQ, 1 × DILK) followed by stretches of predominantly basic amino acids, which matches the typical pattern of the lytic region of *T. infestans* trialysin (Appendix A). Moreover, peptides from different regions of g.1038.t1 (Phor_Comb_C9529 in the transcriptome) were detected only in protein bands below 15 kDa (Appendix A). We thus hypothesized that these proteins evolved multiple lytic regions through domain duplications, resulting in several post-translationally liberated toxins. Therefore, one and six additional putative lytic peptides (peptides 1–6 and 8) were synthesized from g1037.t1 and g1038.t1, respectively. To test whether redulysins from predatory and herbivorous true bugs have different activities, one peptide was synthesized based on a redulysin homolog, which we detected in the salivary gland transcriptome of the phytozoophagous species *Lygus rugulipennis* Poppius (Appendix A). Only two (peptides 9 and 18) and six (peptides 10–15 and 18) synthetic peptides had weak effects on insect cell viability and erythrocyte integrity, respectively. While none of the synthetic peptides inhibited *B. thuringiensis* growth, five and nine peptides had strong inhibitory effects on *E. coli* and *B. subtilis* growth, respectively, at concentrations of 10 µM. Peptide 17 was only active on *E. coli* and peptides 9, 11, and 16 were only on *B. subtilis* (Figure 5B). Dose–response curves for *E. coli* were generated with four peptides that showed strong toxicity below 10 µM. Peptide 12, which corresponds to the lytic region of redulysin g2022.t1, exhibited the most potent effects with an ED50 of <1 µM after 14 h (Figure 5C). Calcium imaging of *D. melanogaster* olfactory neurons treated with 100 µM of peptide 18, which had the strongest effects on insect and red blood cells, caused a slight increase in fluorescence within 10 min. After 10 min, the fluorescence intensity was significantly higher than in the control flies (23.8% compared to −5.2% on average, respectively; Figure 5D). In contrast, peptides 7, 10, and 12, which had strong antimicrobial activity, had no effects on *D. melanogaster* neurons (Appendix A). None of the peptides corresponding to the hypothetically duplicated lytic regions (peptides 1–6, 8) had (negative) effects on insect cell viability, erythrocyte integrity, or microbial growth (Figure 5B).

### 2.4. Activity of Recombinant Venom Protein Family 2 Protein

Recombinant Vpf2 significantly reduced insect cell viability to approximately 70% at concentrations of 0.7 mg/mL, but the toxic effects did not occur when denatured Vpf2 was used (Figure 6A). In addition, the recombinant protein did not affect [Ca^2+^]_i_ of *D. melanogaster* olfactory sensory neurons at concentrations of 0.15 mg/mL (Figure 6B).

## 3. Discussion

Assassin bugs secrete complex venoms that are toxic to a variety of different cell types [12,13,14]. PMG venom from the African assassin bug *P. horrida* facilitates predation and defense and has strong insecticidal and cytotoxic activity [7]. However, the compounds responsible for these effects are yet unknown. We fractionated PMG venom using cation-exchange chromatography and conducted bioassays to identify the cytotoxic compounds. The toxicity against bacterial, insect, and red blood cells was limited to two fractions (Figure 2 and Figure 3), which both contained gelsolin, redulysins, S1 family peptidases, and uncharacterized proteins from the heteropteran venom protein family 2 (Figure 4). The toxic effects could be partly reproduced by synthetic redulysin peptides (Figure 5) and recombinant Vpf2 (Figure 6A), suggesting that both protein families contribute to the cytotoxicity of *P. horrida* PMG venom.

The redulysins are a family of abundant and strongly expressed proteins in the PMG of predatory true bugs [1,7,12,15]. Their homology to the pore-forming protein trialysin from *T. infestans* suggests a broad cytolytic activity, as observed in various assassin bug species [5,7,11,13,14,15]. Trialysin is activated by cleavage of the acidic proregion at the Asp-Glu-Glu-Arg (DEER) site, resulting in the mature peptide that consists of a basic, lysine-rich lytic region, and a non-lytic region [17] (Figure 5A). We identified redulysins in the two cytotoxic fractions and expected that they contribute to the effects on bacterial, insect, and mammalian cells. Several synthetic redulysin peptides had antimicrobial activity against *E. coli* and/or *B. subtilis*, including the peptide from *L. rugulipennis* (Figure 5B,C). The specificity of some peptides to only *E. coli* or *B. subtilis* indicates that *P. horrida* secretes multiple toxic polypeptides directed against different organisms. Such differential toxicity has been described for example in scorpions of the genus *Mesobuthus*, which secrete multiple peptides targeting different bacterial and fungal species [20]. Since trialysins/redulysins are probably stored in the salivary glands as inactive propeptides and are activated only upon injection [11,17], it is unlikely that they play a role in keeping the glands sterile as previously suggested [11,12]. More likely, they act as antimicrobial agents to suppress the growth of microorganisms from prey or plant tissue, preventing the ingestion of potential pathogens. Only some *P. horrida* redulysin peptides showed weak toxicity towards insect and mammalian cells (Figure 5B), but this may not reflect the actual activity range of the native redulysins. In nature, the active peptides consist of the basic lysine-rich region and a non-lytic C-terminal region [11,17] that was omitted in our synthetic peptides. For example, the cytolytic activity of synthetic trialysin peptides was reduced up to 100-fold compared with the native protein [11]. Martins, et al. [21] found considerable differences in toxicity and target-specificity between synthetic peptides representing different portions of the lytic region of trialysin. Similarly, the synthetic fractions of the antimicrobial checacin from the pseudoscorpion *Chelifer cancroides* (L.) exhibited significantly reduced antimicrobial activity compared with the full-length peptide, probably due to a reduced positive net charge [22]. It is therefore possible that the mature full-length *P. horrida* redulysins also act on insect and mammalian cells and contribute to liquefaction, paralysis, and pain induction, despite the weak toxicity of the synthetic peptides. The synthetic peptide corresponding to the redulysin from the phytozoophagous *L. rugulipennis* exhibited the strongest cytotoxic effects against insect and red blood cells (Figure 5B). Although *L. rugulipennis* occasionally feeds on other soft-bodied arthropods [23,24,25], we doubt that the insecticidal activity of its redulysin has an ecological function in this opportunistic predatory behavior. It is more likely a non-target effect of a general cytolytic activity, whose main function is to sterilize food. However, further studies on the ecology and feeding behavior of *L. rugulipennis*, and the mode of action of the native redulysin are necessary to confirm this assumption. Due to their unusually high molecular weight (Appendix A) and redundancies in the amino acid sequences, we hypothesized that the *P. horrida* redulysins g1037.t1 and g1038.t1 might be cleaved multiple times upon activation to produce several active peptides (Appendix A). Such a multidomain multiproduct protein has been found in the venom of the banded Gila monster *Heloderma suspectum cinctum* (Bogert & Martín Del Campo), which expresses several post-translationally liberated toxins encoded by a single mRNA [26]. However, none of the redulysin peptides representing these additional domains (peptides 1–6 and 8) conferred cytotoxic activity toward any of the tested cell types (Figure 5B). Further research should clarify the function, post-translational activation, and activity spectra of these large redulysins. If they are multidomain multiproduct proteins, they might be active against organisms not covered in this study, such as protozoans, fungi, or other bacterial species.

The strong and broad cytotoxicity of *P. horrida* PMG venom indicates that more than one protein family is involved. The uncharacterized heteropteran venom protein family 2 is diverse and abundant in predatory true bugs [1,7,14]. Walker, et al. [19] described a weak homology to trialysins/redulysins and hypothesized that both protein families originated from the same ancestral gene [19]. Venom family 2 proteins from *P. horrida* PMG venom contain 6–10 cysteine residues in the mature peptide after signal peptide cleavage, similar to the non-lytic region of redulysins (Appendix A). Due to their high PMG-specific expression and their presence in the cytotoxic fractions (Figure 4), we hypothesized that family 2 proteins contribute to the venom’s toxicity. Although many different expression systems were tested, we were unable to produce recombinant family 2 proteins in insect cells, bacterial cells, or cell-free systems. The main problems included low cell viability upon induction of expression as well as low protein yields and solubility, indicating a cytotoxic nature of the candidate proteins. Finally, the heterologous expression of one selected family 2 protein, Vpf2, in CHO cells yielded sufficient amounts of soluble protein. Vpf2 was toxic towards insect cells (Figure 6A) but not active against *E. coli* or red blood cells, suggesting that its main ecological role involves prey overwhelming, killing, and/or liquefaction. Venom protein family 2 is a highly diverse protein family with at least 13 genes in *P. horrida* [7], and we assume that other family 2 proteins may have stronger effects or act on other cell types. Moreover, we cannot exclude that Vpf2 acts synergistically with other venom compounds, thereby increasing its potency and expanding the target range. Future research should therefore focus on the activity of different venom protein family 2 proteins from different species, also in combination with other venom compounds, to determine their mode of action and ecological function.

In addition to cell-culture-based assays, the effects of *P. horrida* venom on [Ca^2+^]_i_ of a subset of neurons in the brain of *D. melanogaster* were tested. Calcium ions play an important role in neuronal signaling, muscle contraction, and apoptosis [27]. Moreover, calcium influx is a common indicator of pain, because it is a crucial step in the activation of nociceptive neurons [14]. Several studies demonstrated the rapid increase of [Ca^2+^]_i_ in mammalian sensory neurons upon treatment with defensive venom secretions [14,28,29]. Most likely, these effects are due to the disruption and depolarization of membranes by pain-inducing lytic proteins [14]. We observed a highly complex pattern of fluctuations in [Ca^2+^]_i_ upon treatment with unfractionated venom (Figure 3C), indicating the presence of several compounds with different kinetics and modes of action. First, the relative change in fluorescence intensity dropped down to approx. −50% in less than a minute, probably representing the effects of fast-acting compounds such as neurotoxins. Venom peptides from assassin bugs that have been suggested to act as neurotoxins include ptu1-like peptides, cystatins, and CUB domain proteins [16,30]. The fast decrease in [Ca^2+^]_i_ was followed by a gradual increase over several minutes. The subsequent rapid [Ca^2+^]_i_ drop was followed by a sharp increase that can probably be explained by a complete breakdown of membrane potential due to pore-forming proteins, leading to the death of the fly. The fluorescence intensity then gradually decreased, probably due to leakage of the fluorescent dye through membrane pores [14,31]. The complex effects could not be fully recreated by applying the cytolytic fractions A or B, thus indicating that other non-lytic compounds not present in these fractions contribute to the venom’s insecticidal activity. Application of fraction A or B did not lead to an increase of [Ca^2+^]_i_ as would be typical for pore-forming proteins [14,31], but instead resulted in a rapid decrease in most replicates (Figure 3D and Appendix A). LC-MS/MS revealed the presence of gelsolin in both fractions (Figure 4), a calcium-binding protein that is associated with actin depolymerization in prey animals [32,33]. We assume that large amounts of extracellular calcium were bound by gelsolin after the application of the fractions, resulting in a change in the concentration gradient between intracellular and extracellular media. Under these conditions, the action of pore-forming proteins such as redulysins would lead to a decrease of [Ca^2+^]_i_ to balance the gradient. Application of the synthetic *L. rugulipennis* redulysin peptide resulted in a slight increase in [Ca^2+^]_i_ (Figure 5D), thus confirming its presumed pore-forming activity. Although other tested redulysin peptides from *P. horrida* showed no effects on [Ca^2+^]_i_ (Appendix A), we assume that they generally share a mode of action with the *L. rugulipennis* redulysin. The effects of the synthetic *L. rugulipennis* redulysin on [Ca^2+^]_i_ in *D. melanogaster* brains were observed at concentrations 10-fold higher than the toxic effects on cultured insect cells, thus indicating that redulysins may rely on other compounds, supporting translocation to their target sites in more complex (i.e., in vivo) environments. The central nervous system of *D. melanogaster* consists of 10% glial cells, which form a 2–3 µm thick layer around the brain and fulfill a variety of functions, including the generation of a blood-brain barrier to provide chemoprotection [34,35]. These protective glial cells likely impede the action of redulysins on *D. melanogaster* neuronal cells. An important so-called “venom spreading factor” found in many venoms is hyaluronidase, a carbohydrase that degrades polysaccharides in the extracellular matrix of animals and therefore facilitates the rapid spread of venom molecules to reach their site of action [10,36,37]. *Psytalla horrida* venom also contains hyaluronidase [7] and we hypothesize that it enhances the potency of cytotoxic peptides such as redulysins. Future experiments should include combinations of redulysin peptides with potential spreading factors such as hyaluronidase or proteases to verify whether they act synergistically. Similarly, Vpf2 should be tested in combination with putative spreading factors to further investigate its role as an insecticidal peptide.

## 4. Materials and Methods

### 4.1. Insects

*P. horrida* specimens were obtained from an insectarium breeding source (Jörg Bernhardt, personal communication). They were kept at room temperature in terraria containing sand, coconut fibers, and bark and fed once per week with *Gryllus assimilis* (Fabricius), which were obtained from Tropic Shop (Nordhorn, Germany).

*L. rugulipennis* specimens were obtained from Katz Biotech AG (Baruth/Mark, Germany) and used directly for RNAseq.

For calcium imaging, 5–7 days old *D. melanogaster* flies expressing GCaMP6s in olfactory sensory neurons were used (genotype: +/+; UAS-GCaMP6s/UAS-GCaMP6s; Orco-Gal4/Orco-Gal4 (from Ilona Grunwald Kadow, Silke Trautheim, MPI-CE Jena, Germany)). The flies were reared in plastic vials at 12 h/12 h light/dark cycle at 25 °C and 70% humidity. Every two weeks, they were transferred into clean vials containing a fresh diet (standard cornmeal medium: sugar beet molasses, beer yeast, powdered agar, polenta, propionic acid, and Nipagin and water).

### 4.2. Venom Extraction

Venom was extracted from the PMG of fifth-instar or adult assassin bugs. Individuals were anesthetized at −20 °C for 5 min before dissection in phosphate-buffered saline (PBS). The posterior lobe was separated from the AMG and AG and transferred to a pre-cooled tube containing 100 µL of 20 mM MES (pH 5.5) on ice. After briefly vortexing, the samples were centrifuged (4000× *g*, 3.5 min at 4 °C), and the supernatant was further clarified by additional centrifugation (13,000× *g*, 3 min, 4 °C). The venom extracts of several individuals were pooled and stored at −20 °C for analysis. The total protein concentration in the venom samples was determined using an N60 Nanophotometer (Implen, Munich, Germany).

### 4.3. Venom Fractionation

PMG venom extracts were fractionated by fast protein liquid chromatography (FPLC). Then, 2 mL of venom (7.6 mg/mL) in 20 mM MES at pH 5.5 (buffer A) were filtered through a 0.22 μm syringe filter and loaded onto a 1 mL RESOURCE S cation-exchange column (GE Healthcare, Chicago, IL, USA), which was pre-equilibrated with buffer A. Chromatography was carried out at a flow rate of 0.5 mL/min. After initial elution with buffer A, the salt concentration was gradually increased by adding 20 mM MES containing 1 M NaCl at pH 5.5 (buffer B) up to a maximum salt concentration of 600 mM NaCl (=60% buffer B). In total, 43 protein-containing fractions were collected and stored at −20 °C for analysis.

### 4.4. Gel Electrophoresis and LC-MS/MS

Sodium dodecylsulfate polyacrylamide gel electrophoresis (SDS-PAGE) was performed using 4–12% Criterion XT gradient gels (BioRad, Hercules, CA, USA) with XT MES running buffer (BioRad, Hercules, CA, USA). Gels were run at 125 V for 1 h along with a pre-stained protein standard (Thermo Fisher Scientific, Waltham, MA, USA) and stained for 1.5 h in a 1:1 (*v/v*) mixture of Coomassie Brilliant Blue R-250 and colloidal Coomassie Brilliant Blue G-250 (Thermo Fisher Scientific, Waltham, MA, USA). After washing in Millipore water overnight to remove excess dye, the stained gels were scanned and analyzed. From selected gel lanes, the protein bands were excised for tryptic digestion [38] and reconstituted in 50 μL of 1% formic acid in water. Liquid chromatography–tandem mass spectrometry (LC-MS/MS) and subsequent data collection were performed as described previously [7].

### 4.5. Proteomic Data Processing and Protein Identification

The ProteinLynx Global Server (PLGS) v2.5.2 (Waters, Milford, MA, United States) was used to process DDA raw data. The spectra were screened against a sub-database of common contaminants including human keratins and trypsin. Unmatched spectra were interpreted de novo to yield peptide sequences for homology-based searching using MS BLAST [39] installed on a local server. We carried out MS BLAST searches against the Arthropoda database (downloaded from NCBI on 12 February 2019) and the *P. horrida* sub-database obtained from in silico translation of the transcriptome generated by Fischer, et al. [7]. In addition, the pkl files generated from raw data were searched against the NCBInr database (downloaded on 12 February 2019) combined with the *P. horrida* sub-database using MASCOT v2.6.0. The following parameters were used: fixed precursor ion mass tolerance of 15 ppm for survey peptide, fragment ion mass tolerance of 0.1 Da, 1 missed cleavage, fixed carbamidomethylation of cysteines, and possible oxidation of methionine.

### 4.6. RNAseq of Lygus Rugulipennis

The salivary gland complex (SG), gut, and remaining body tissue of adult *L. rugulipennis* specimens were dissected in PBS and transferred into separate ceramic bead tubes containing 500 µL of TRI Reagent (Sigma-Aldrich, St. Loius, MO, USA). The tissues of six individuals were pooled and homogenized using a TissueLyser LT (Qiagen, Venlo, Netherlands). The Direct-zol RNA Miniprep Kit was used to extract total RNA according to the manufacturer’s instructions (Zymo Research, Irvine, United States). RNA quantity and integrity were determined using an N60 nanophotometer and an Agilent 2100 Bioanalyzer and RNA Nanochip (Agilent Technologies, Santa Clara, CA, USA), respectively.

The SG, gut and remaining body tissue transcriptomes were sequenced by the Max-Planck Genome Center Cologne (http://mpgc.mpipz.mpg.de/home/(accessed on 19 April 2023)) using an Illumina HiSeq3000 Genome Analyzer platform. Poly-A mRNA was isolated from 1 µg total RNA using oligo-dT attached to magnetic beads. The RNA was then fragmented to an average of 250 bp and sequencing libraries were generated using the TruSeq RNA library preparation kit. Paired-end (2 × 150 bp) read technology was used for sequencing. Detailed information on sequencing and assembly is summarized in Appendix A. An in-house assembly and annotation pipeline, described by Fischer et al. [7], was used to process the generated reads. Briefly, quality control and transcriptome assemblies were performed using CLC Genomics Workbench v11.1. The presumed optimal consensus transcriptome was selected, as previously described [40]. Annotations were added using BLAST, Gene Ontology, and InterProScan in OmicsBox (https://www.biobam.com/omicsbox, Valencia, Spain) as described by [41]. Transcriptome completeness was assessed using the Benchmarking Universal Single Copy Orthologs (BUSCO) v3 pipeline [42]. Digital gene expression analysis was conducted in CLC Genomics Workbench v11.1, using previously described parameters for read mapping and normalization [43].

### 4.7. Psytalla Horrida Genome Assembly and Identification of Redulysin-Encoding Candidate Genes

High molecular weight (HMW) genomic DNA was isolated from a single adult *P. horrida* individual with the Nanobind tissue kit (Circulomics, Baltimore, MD, USA). Further processing of the isolated DNA was performed with the Short Read Eliminator SRE kit (Circulomics, Baltimore, MD, USA) to enrich for fragments longer than 25 kb. DNA purity was assessed using a NanoPhotometer P330 (Implen, Munich, Germany) and DNA quantity using a Qubit dsDNA BR assay kit in combination with a Qubit 2.0 Fluorometer (Invitrogen). The size-selected HMW DNA was used as the starting material for the preparation of ONT libraries, following the manufacturer’s guidelines for the Ligation Sequencing Kit SQK-LSK110 (Oxford Nanopore Technologies, Oxford, UK), and sequenced on R9.4.1 flow cells using the MinION sequencing device. High-accuracy basecalling was performed using GUPPY version 4.0.11. Flye (v2.8) assembler was used to generate a de novo genome assembly from the ONT data, followed by four iterations of polishing using Racon, and one round of error correction using Medaka. For final error correction of the ONT-based assembly, we used ntEdit with 20× coverage paired-end (2 × 150 bp) Illumina data generated from the same HMW DNA used for ONT sequencing. To remove duplications (heterozygous regions) and generate a haploid genome for further downstream analysis, we used purge_haplotigs. Genome annotation was done using the Augustus gene prediction tool and candidate redulysin genes were identified with Blast search tools implemented in Geneious Prime (v2022.0.2), using the *P. horrida* transcriptome-derived sequences.

### 4.8. Synthesis of Redulysin Peptides

Potential lytic regions of *P. horrida* redulysins were synthesized by Genscript Biotech Corporation (Piscataway Township, NJ, USA) using solid-phase synthesis. Lyophilized peptides were reconstituted in 20 mM MES pH 5.5 to obtain stock solutions of 1 mM, which were stored at −20 °C.

### 4.9. Heterologous Expression of Venom Protein Family 2 Proteins

One venom protein family 2 protein (Phor_Comb_C31143; hereafter Vpf2) homologous to venom protein family 2 protein 4 from *P. plagipennis* (Genbank accession: AQM58361.1) was heterologously expressed in CHO cells by Genscript Biotech Corporation. The recombinant proteins were purified from cell culture supernatants using HisTrap™ FF Crude+ HiLoad™ 26/600 Superdex. Proteins were eluted in PBS buffer (pH 7.2), purity was confirmed with SDS-PAGE under non-reducing conditions (≥95%) and SEC-HPLC (>99%), and purified proteins were stored at −80 °C.

### 4.10. Cell Viability Assay

*Sf*9 cells were cultured in sterile 96-well plates (Greiner, Kremsmünster, Austria) using Sf-900 II SFM medium (Gibco) containing 0.05 mg/mL gentamycin. After 24 h, the culture medium was replaced by 100 µL of a 1:10 dilution of venom fraction or dissolved redulysin peptide (100 µM) in *Sf*9 culture medium or a 1:1 dilution of recombinant Vpf2 in *Sf*9 culture medium (*n* = 6) and incubated at 27 °C for 4 h. The positive control was a 0.1% Triton x-100 solution in *Sf*9 (*n* = 6). Negative controls were 20 mM MES (pH 5.5) for synthetic redulysin peptides, 20 mM MES + 0.4 M NaCl (pH 5.5) as well as 20 mM MES + 1 M NaCl (pH 5.5), for fractions A and B, respectively (*n* = 6). For the cell viability assay with recombinant Vpf2, we used PBS and denatured protein (heated at 95 °C for 10 min) as negative controls. After incubation, the culture medium was replaced with a medium containing 0.5 mg/mL thiazolyl blue tetrazolium bromide (MTT, Sigma Aldrich, St. Loius, MO, USA). After incubation at 27 °C for 2 h, the MTT mixture was removed and 50 µL of dimethyl sulfoxide (DMSO, Sigma Aldrich, St. Loius, MO, USA) was added. The plates were incubated at 27 °C for 10 min, briefly vortexed and the absorbance at 540 nm was read in an Infinite m200 plate reader (Tecan, Männedorf, Switzerland). The average absorbance value of pure DMSO was subtracted from all values and the relative cell viability was calculated in relation to the negative control (defined as 100%).

### 4.11. Bacterial Growth Inhibition Assay

Fifty µL of bacterial overnight culture (*E. coli*, *B. subtilis* or *B. thuringiensis*) in lysogeny broth (LB) medium was added to 5 mL LB medium and incubated at 37 °C for 2–3 h. The culture was then diluted with LB medium to an OD600 of 0.003. In a 96-well plate (Greiner), 10 µL of venom fraction or dissolved redulysin peptide (100 µM) was mixed with 90 µL of bacteria dilution (*n* = 3). Similar dilutions were prepared with gentamycin (0.5 mg/mL) as a positive control (*n* = 3). Negative controls were 20 mM MES (pH 5.5) for synthetic redulysin peptides, and 20 mM MES + 0.4 M NaCl (pH 5.5) as well as 20 mM MES + 1 M NaCl (pH 5.5), for fractions A and B, respectively (*n* = 3). The mixtures were incubated at 30 °C in an Infinite m200 plate reader (Tecan, Männedorf, Switzerland) and the absorbance at 595 nm was read in 5 min intervals over a period of 24 h. The average absorbance value of PBS in sterile LB medium (1:10) was subtracted from all values and the relative growth after 12.5 h (OD595 of a PBS control ~ 0.36, log-phase) was calculated in relation to the respective negative control (defined as 100%). For the dose-response analysis, varying concentrations (0, 10, 15, 20, 25, 30, 35, 40, 45, 50, 55, and 60 µM) of selected redulysin peptides were tested (*n* = 2) and the OD595 values after 14 h compared.

### 4.12. Hemolysis Assay

Erythrocytes were harvested by centrifuging defibrinated horse blood (Thermo Fisher Scientific) at 1500 g for 3 min. The cells were washed three times with PBS and a 1:10 erythrocyte suspension in PBS was prepared. 20 µL of venom fraction or dissolved redulysin peptide (100 µM) were mixed with 180 µL cell suspension (*n* = 3) in a 96-well plate and incubated at 37 °C for 1 h. The positive control was 1% Triton x-100 (*n* = 3). Negative controls were 20 mM MES (pH 5.5) for synthetic redulysin peptides, and 20 mM MES + 0.4 M NaCl (pH 5.5) as well as 20 mM MES + 1 M NaCl (pH 5.5), for fractions A and B, respectively (*n* = 3). After incubation, the cell suspensions were centrifuged at 2000× *g* for 10 min and the supernatants were transferred into a new clear 96-well plate. The absorbance at 440 nm was read in an Infinite m200 plate reader (Tecan, Männedorf, Switzerland). The average absorbance value of the negative control was subtracted from all values and the relative cell integrity was calculated in relation to the negative control (defined as 100%).

### 4.13. Dissection of Flies for Calcium Imaging

*D. melanogaster* females were anesthetized on ice before dissection. The flies were immobilized by pushing the cervical region into a slit of a copper plate (Athene slot diaphragm, 125-µm slot, Plano, Wetzlar, Germany) that was glued to a mounting stage made of Plexiglas. A fine needle (minutiens 0.10 mm, Austerlitz Insect Pins, Slavkov u Brna, Czech Republic) was pressed on top of the proboscis and fixed with beeswax on both sides of the mounting stage in order to immobilize the head. Furthermore, the backside of the head was glued to the copper plate by using 3-component dentist glue (Protemp™ II, 3M). Both ends of a fine tetrode wire (Redi Ohm 800, H.P. Reid Inc. Palm Coast, FL, USA) were fixed to a plastic plate with beeswax. The wire was inserted into the ptilinal suture. Using screws in the mounting stage, the plastic plate with the attached wire was slowly pushed forward to displace the antennae slightly to the front. A plastic plate with a hole was glued to a polyethylene foil and an additional hole with a smaller diameter was punched through the foil. The plate was placed on top of the fly’s head so that the hole in the foil exposed the head. 2-component silicon (Kwik-Sil™, World Precision Instruments, Sarasota, FL, USA) was used to seal the space between the edges of the foil and the fly’s head capsule. Then, a droplet of Ringer’s solution (NaCl: 130 mM, KCl: 5 mM, MgCl_2_: 2 mM, CaCl_2_: 2 mM, Sucrose: 36 mM, HEPES-NaOH (pH 7.3): 5 mM) was added onto the head. The head capsule was opened dorsally with a fine scalpel (Micro Knive, Fine Science Tools, Heidelberg, Germany), and fat body tissue, glands, and tracheae were removed with fine forceps to ensure optical access to the brain.

### 4.14. Calcium Imaging

In order to monitor calcium signals in the fly brain, we functionally imaged calcium changes in the antennal lobe, the first olfactory neuropil of insects, with and without venom application at a 2-photon microscope. Before the experiment, the droplet of Ringer’s solution added during the dissection was removed with tissue paper and replaced with 40 µL Ringer’s solution. The dissected fly was placed on the stage of a ZEISS LSM 710 NLO confocal microscope (Carl Zeiss, Jena, Germany) equipped with an infrared Chameleon Ultra diode-pumped laser (Coherent, Santa Clara, CA, USA). The ZEN software (Carl Zeiss, Jena, Germany) was used to control the microscope. The fluorophore of the expressed GCaMP6s was excited using a laser wavelength of 925 nm. A 63× water immersion objective (W Plan-Apochromat 63×/1.0 VIS-IR, Zeiss, Carl Zeiss, Jena, Germany) was used to visualize a plane of the right antennal lobe containing the DM1 and DM2 glomeruli. The laser power was set between 10% and 30%, depending on the expression level of GCaMP6s in individual flies and the master gain was set to 700. The frame size of the acquired time series was 248 × 250 pixels and the frame rate 4 Hz.

In order to confirm that the dissected flies were alive and the brains undamaged, their response to an odor (3-hexanone at a concentration of 10-2) was tested before the experiments. An electronically controlled odor delivery system consisting of flexible Teflon tubes guiding two converging airstreams (0.5 L/min each) to the antennae was used. A solenoid valve controlled by the LabVIEW software (National Instruments, Austin, TX, USA) was installed in one of these airstreams and switched between a tube transporting pure air and a tube that entered a 50 mL glass bottle (Schott, Jena, Germany) containing 1 mL of a diluted odorant. A time series of 10 s was acquired, in which the odor pulse began after 2 s and lasted for 2 s. After confirming that the flies were alive, a time series of 15 s was acquired in order to determine the GCaMP6s base fluorescence. Subsequently, 5 µL of unfractionated PMG venom (18 mg/mL in MES buffer), venom fractions, redulysin peptides (900 µM), or recombinant Vpf2 (1.4. mg/mL) were added to the ringer solution and a time series of 10 min was started immediately afterward. PBS as well as 20 mM MES (pH 5.5) and 20 mM MES containing 0.6 M NaCl (pH 5.5) were used as negative controls, representing the lowest highest possible NaCl concentrations in the fractions (excluding flow-through).

### 4.15. Data Analysis

Data analysis was performed using R v4.0.3 and the integrated development environment RStudio v1.2.1335 [44]. For the hemolysis, cell viability, and bacterial growth inhibition assays, we performed a one-way analysis of variance (ANOVA) with pairwise *t*-tests or Kruskal-Wallis rank sum tests with pairwise Dunn’s tests (Appendix A) using the FSA package [45]. Dose-response curves for *E. coli* treated with redulysin peptides were plotted after fitting the data for each treatment to a logistic model using the drc package [46]. All plots were created using the ggplot2 package [47] or the R built-in plot() function. To analyze calcium imaging scans, the ImageJ software [48,49] was used. The ROI manager was applied to select the antennal lobe and extract the average brightness value of pixels within the selected region (including movements) across the entire time series. The extracted values were used to calculate and plot the change in fluorescence intensity (ΔF/F) in percent. For each individual, the mean value of the 15 s time series representing the base fluorescence was subtracted from the values of the 10 min time series to calculate ΔF. Then, ΔF was divided by the base fluorescence to calculate ΔF/F and multiplied by 100 to obtain ΔF/F [%].

## Figures and Tables

**Figure 1 toxins-15-00302-f001:**
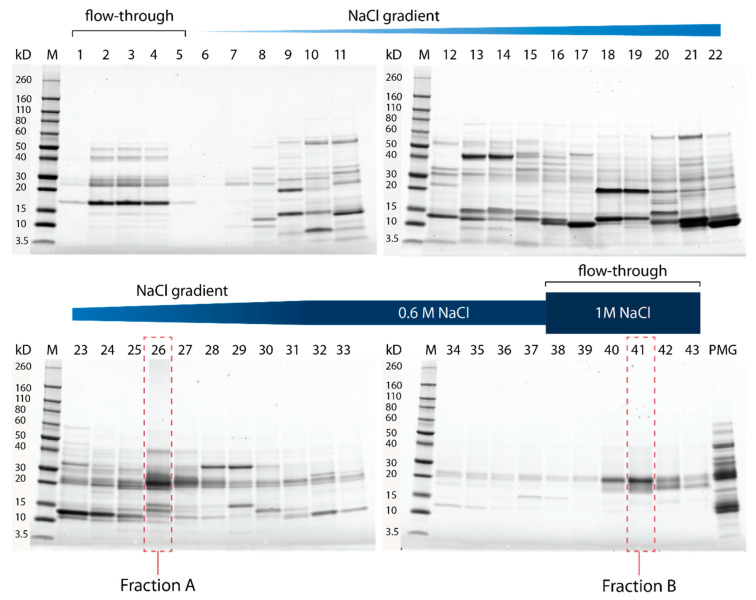
SDS-PAGE analysis of fractions of *Psytalla horrida* PMG venom. 1–43 = venom fractions obtained through cation exchange chromatography; PMG = unfractionated PMG venom; M = protein marker. Fraction A and fraction B are highlighted in red. The first five fractions correspond to the flow-through with buffer A, whereas the last six fractions correspond to the flow-through with buffer B.

**Figure 2 toxins-15-00302-f002:**
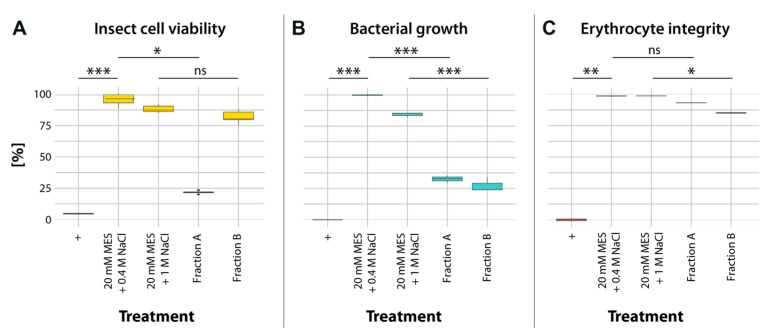
Bioactivity of fraction A and fraction B in comparison with a negative control (20 mM MES + 0.4 M NaCl or 20 mM MES + 1 M NaCl, respectively), and (+) a positive control treatment. Significant differences compared to the respective negative control are highlighted with asterisks (* *p* ≤ 0.05; ** *p* ≤ 0.01; *** *p* ≤ 0.001; ns = not significant). (**A**) Fraction A led to reduced viability of treated sf9 cells; (+) = 0.1% Triton x-100. Statistical test: Dunn’s tests, *n* = 6. (**B**) Both fractions significantly delayed the growth of *Escherichia coli*; (+) = 0.05 mg/mL gentamycin. Statistical test: pairwise *t*-tests, *n* = 3. (**C**) Fraction B caused mild hemolysis of horse erythrocytes; (+) = 0.1% Triton x-100. Statistical test: Dunn’s tests, *n* = 3.

**Figure 3 toxins-15-00302-f003:**
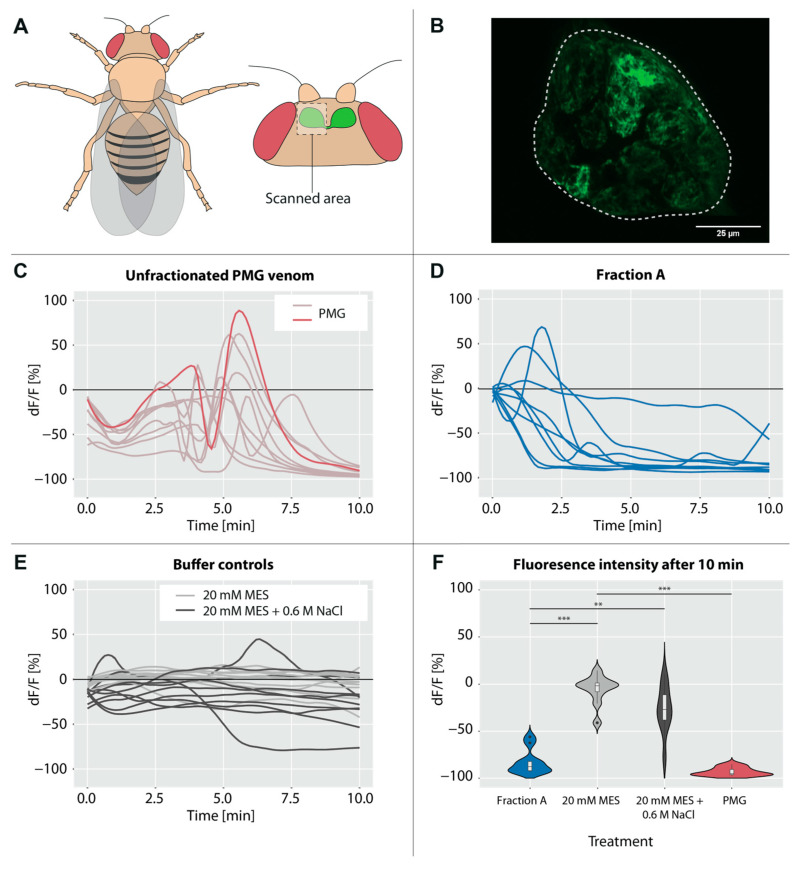
Calcium imaging of *Drosophila melanogaster* antennal lobes using genetic expression of the calcium-sensitive protein GCaMP6s. (**A**) Schematic of *Drosophila melanogaster* highlighting the left antennal lobe that was imaged. (**B**) Representative image of an antennal lobe with GCaMP6s expression in olfactory sensory neurons without venom treatment. The dashed line represents the region selected with the ROI manager to extract the brightness values. (**C**) Changes of dF/F (representing fluorescence changes as an indicator for intracellular calcium concentration) after treatment with unfractionated PMG venom; for highlighting the observed representative course of the fluorescence change, one replicate is marked in dark red. (**D**) Changes of dF/F after treatment with fraction A. (**E**) Changes of dF/F after treatment with 20 mM MES and 20 mM MES + 0.6 M NaCl (negative controls). (**F**) Changes in fluorescence intensity after 10 min. Boxplots within the violin plots represent the median (line), interquartile range (box), and data range (whiskers). Significant differences between treatments are highlighted with asterisks (** *p* ≤ 0.01; *** *p* ≤ 0.001, Dunn’s tests, *n* = 10 (PMG, fraction A, 20 mM MES + 0.6 M NaCl), *n* = 11 (20 mM MES)).

**Figure 4 toxins-15-00302-f004:**
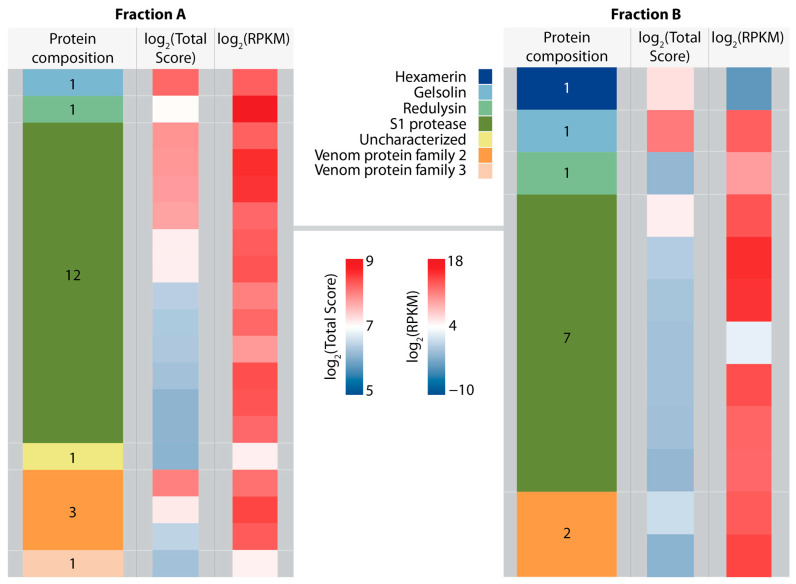
Protein composition of fraction A and fraction B. Color-coded blocks represent the number of contigs identified in transcriptome datasets and verified by proteomic analysis, which encodes specific classes of functional proteins. Log_2_(Total Score) depicts the logarithmic total score of all matched peptides identified by LC-MS/MS. Log_2_(RPKM) shows the expression level of the respective contig in the PMG.

**Figure 5 toxins-15-00302-f005:**
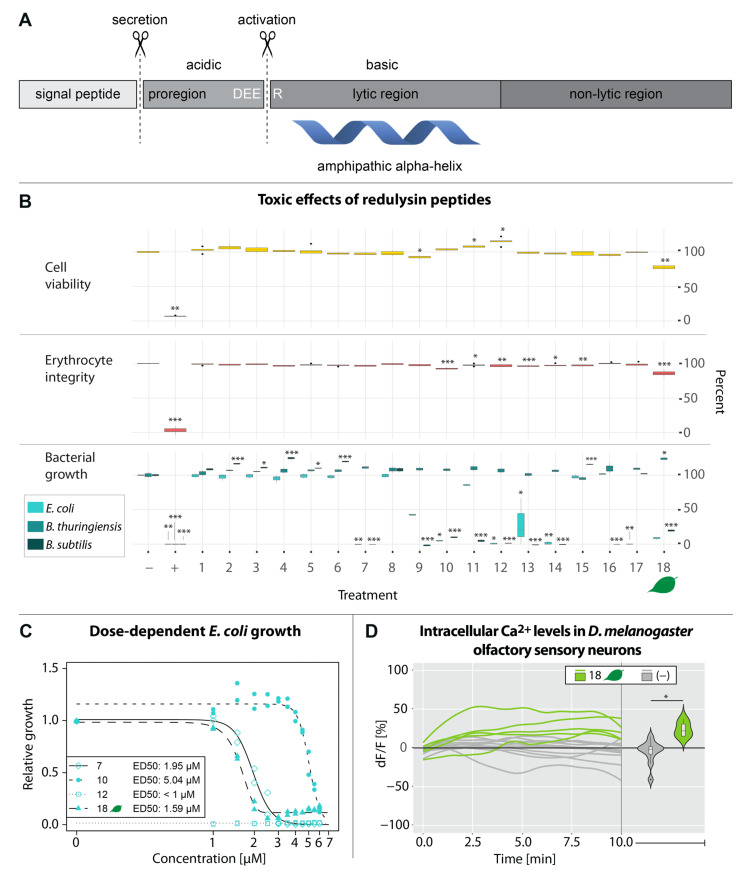
Bioactivity of synthetic redulysin peptides. (**A**) The structure of redulysins consists of a signal peptide, a proregion, a lytic region, and a non-lytic region. After cleavage of the proregion at the DEER motif, the amphipathic pore-forming alpha-helix is exposed. (**B**) Insect cell viability, erythrocyte integrity and growth of *Escherichia coli, Bacillus subtilis* and *Bacillus thuringiensis* in presence of 10 µM redulysin peptide; (−) = 20 mM MES pH 5.5 (=100%), (+) = 0.5 mg/mL gentamycin or 0.1% triton-x 100. Significant differences compared to the negative control are highlighted with asterisks (* *p* ≤ 0.05; ** *p* ≤ 0.01; *** *p* ≤ 0.001, pairwise *t*-tests or Dunn’s tests, *n* = 3). (**C**) Dose-dependent growth of *Escherichia coli* 14 h after treatment with selected redulysin peptides at varying concentrations. The data were fitted to a logistic model and plotted as dose-response curves (*n* = 2). (**D**) Change of intracellular Ca^2+^ levels (represented by dF/F) in *Drosophila melanogaster* olfactory sensory neurons after treatment with 100 µM redulysin peptide 18. Violin plots represent the change in fluorescence intensity after 10 min (* *p* ≤ 0.05, Kruskal–Wallis test). Boxplots within the violin plots represent the median (line), interquartile range (box), and data range (whiskers). (−) = 20 mM MES pH 5.5. Peptide 18, which corresponds to the redulysin from the phytozoophagous bug *Lygus rugulipennis* is marked with a leaf symbol in (**B**–**D**).

**Figure 6 toxins-15-00302-f006:**
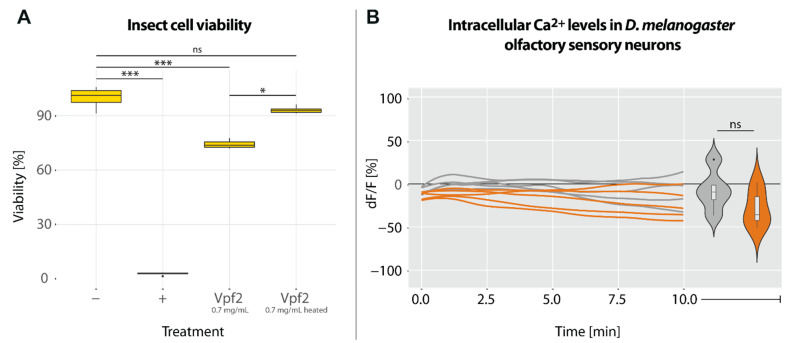
Bioactivity of a recombinant venom protein family 2 (Vpf2) protein. (**A**) Insect cell viability in the presence of 0.7 mg/mL Vpf2 and heated Vpf2; (−) = PBS (=100%), (+) = 0.1% triton-x 100. Significant differences compared to the negative control are highlighted with asterisks (* *p* ≤ 0.05; *** *p* ≤ 0.001; ns = not significant, Dunn’s tests, *n* = 5). (**B**) Course of intracellular Ca2+ levels in *Drosophila melanogaster* olfactory neurons after treatment with 0.2 mg/mL Vpf2 or PBS. Violin plots represent the change in fluorescence intensity after 10 min (ns = not significant; α = 0.05, Kruskal-Wallis test). Boxplots within the violin plots represent the median (line), interquartile range (box), and data range (whiskers).

## Data Availability

The data presented in this study are available in the Appendix A.

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
