# Peer review of "An Assassin’s Secret: Multifunctional Cytotoxic Compounds in the Predation Venom of the Assassin Bug *Psytalla horrida* (Reduviidae, Hemiptera)"

_toxins, 2023, doi:10.3390/toxins15040302_

Round 1
Reviewer 1 Report
The manuscript identified the venom compounds secreted by posterior main gland of the African assassin bug Psytalla horrida, screened the fractions and investigated the function of candidate components by peptide synthesis and activity test, which will contribute to venom function analysis among assassin bugs. While I think there are several aspects of this manuscript that can be improved. Here are my general concerns and suggestions.
1. Species names are not used in a correct way throughout the manuscript. All species names should be in italics. While, the authors provide many scientific names in non-italics, which is not very scientific. For example, line 96 Drosophila melanogaster, line 122 D. melanogaster, line 152 P. horrida, line 158 T. infestans, line 242…etc. Please carefully check and revise.
2. Line 108, “P. horrida” --> “Psytalla horrida”. It is better to use the full scientific name in figure captions. Similar cases exist in line 118, line 122…etc. Please carefully check and revise.
3. The authors extracted PMG venom from the salivary glands of fifth-instar or adult assassin bugs and use the extracts for analysis. I am wondering if the PMG venom composition from the nymphs is exactly same as the adults? Is there any evidence that the PMG extracts between the nymphs and adults are similar? Based on naturalistic observations, adult assassin bugs are more aggressive and better at killing the preys than the immatures, which may imply their differences in venom composition.
4. In order to test whether redulysins from predatory and herbivorous true bugs have different activities, the authors chose Lygus rugulipennis as the phytophagous representative and selected a redulysin homolog based on its salivary gland transcriptome. However, L. rugulipennis is an omnivore feeding on both plants and prey, like many other mirids (Coll & Guershon, 2002). Thus, I think it is not appropriate to use the redulysin homolog screened from L. rugulipennis as a phytophagous comparison.
5. Redulysin and venom protein family 2 (Vpf2) are two most important components in venom of assassin bugs and were hypothesized to contribute to the venom’s toxicity. But different from redulysin, Vpf2 was proved to be toxic towards insect cells but not active against E. coli or red blood cells in present study, which I think is interesting. Is it possibly because that Vpf2 needs to perform function by interacting with other toxic peptides?
Reviewer 2 Report
The authors propose to identify the compounds of the venom of the African killer bug P. horrida responsible for antimicrobial, hemolytic and insecticidal (cytotoxic) effects.
This raw venom was tested and then fractionated. Two fractions A and B were screened for toxic effects. The composition of these active fractions was studied by LC-MS/MS and some peptides were selected and produced, either by peptide synthesis or by heterologous expression. The authors report that some synthetic peptides are antibacterial and that several heterologous proteins exert toxic activity on SF9 insect cells.
The article is well constructed and well written. It is easy to read. The supplementary data provide some answers to the questions. The experiments performed are relevant and diversified. I have no particular reproach to make: some of the synthesized peptides clearly appear as anti-bacterial. On the other hand, the authors indicate that some heterologous proteins are significantly cytotoxic (figure 6): if the statistics indicate a significant difference with the control, it seems to me that with 70% viability the cytotoxicity is low, especially since the test was performed at 700µg/mL. This seems enormous to me.
Finally, the article does not really demonstrate that the synthetic molecules tested are at the origin of the cytotoxicity of the venom. Knowing that fractions A and B are still complex (figure 1), are there not other molecules in this venom that have not been tested and that could be the cause of the observed effects?
To conclude, even if the authors did not demonstrate to have clearly identified the active molecules of this venom, this article brings however some interesting general knowledge on the nature of the P. horrida bug venom. For this reason, I have no opposition to the publication of the article in its present form.
Reviewer 3 Report
Venom are sources of novel chemical entities with promising therapeutic applications. They are also valuable compounds to study the ecology and evolution of venomous animals. In this sense, this research article is an interesting study that brings novel insights into biochemistry and biological effects of Psytalla horrida venom.
1. Some sentences should be supported by references. Please double check the lines (31-32: recent studies, which?; 42-43, 55-57, 74-75, and so on) and the entire manuscript.
2. Lines 118-119, 152, 154, 193, 208 and so on. Scientific names are always italicised.
3. Authors should standardise the graphs and colours. Controls and fractions are represented using different colours in different figures. For example, in figure 2 fraction A is green and in figure 3 fraction A is blue.
4. Why the effect of crude venom was not analysed? It would be interesting include the effect of whole venom in figure 2.
5. Figure 3B. Please include bar length.
6. Figure 5 B. This representation is uncommon for antibacterial effect. Why the authors have not assessed the MIC?
7. I recommend the authors to use in silico approaches to corroborate the haemolytic effect of peptides. This additional evaluation contributes to expand the use of in silico methods and assess their potential. Please also double check the stastitical analysis shown in Figure 5B.
Round 2
Reviewer 1 Report
The authors have carefully revised the manuscript and series changes have been made to improve the main text. Now I think the paper is appropriate to be published in present form.